# Study of Organic Matter of Unconventional Reservoirs by IR Spectroscopy and IR Microscopy

**Natalya Tanykova [1], Yuliya Petrova [1],\*, Julia Kostina [2], Elena Kozlova [3], Evgenia Leushina [3] and Mikhail Spasennykh [1,3]**

1    Institute of Natural and Technical Sciences, Surgut State University, 1 Lenina Prospect, 628412 Surgut, Russia; tanykova_ng@surgu.ru (N.T.); m.spasennykh@skoltech.ru (M.S.)

2    A.V.Topchiev Institute of Petrochemical Synthesis, Russian Academy of Sciences, Leninsky Prospect 29, 119991 Moscow, Russia; julia@ips.ac.ru

3    Center for Hydrocarbon Recovery, Skolkovo Institute of Science and Technology, Bolshoy Boulevard 30, bld. 1, 121205 Moscow, Russia; E.Kozlova@skoltech.ru (E.K.); E.Leushina@skoltech.ru (E.L.)

\*    Correspondence: petrova_juju@surgu.ru; Tel.: +7-3462-76-3091

**Abstract:** The study of organic matter content and composition in source rocks using the methods of organic geochemistry is an important part of unconventional reservoir characterization. The aim of this work was the structural group analysis of organic matter directly in the source rock in combination with a quantitative assessment and surface distribution analysis of the rock sample by FTIR spectroscopy and FTIR microscopy. We have developed new experimental procedures for semi-quantitative assessment of the organic matter content, composition and distribution in the source rocks and applied these procedures for the study of the samples from the Bazhenov shale formation (West Siberia, Russia). The results have been verified using the data from the study of organic matter obtained by Rock-Eval pyrolysis and differential thermal analysis. The obtained results demonstrate the prospects of FTIR spectroscopy and FTIR microscopy application for non-destructive and express analysis of the chemical structure and distribution of organic matter in rocks.

**Keywords:** FTIR spectroscopy; FTIR microscopy; organic matter; kerogen; oil shale; rock

## 1. Introduction

Mineralogical and geochemical information on reservoirs, including unconventional, and different types of source rocks is necessary to assess the gas and oil generation potential and production technologies. The standard methods in the petroleum industry for obtaining these properties are bulk measurements on crushed rock samples and they can take several hours to perform. To deepen the understanding of oil forming processes, it is important to develop the methods for express and reliable analysis of mineralogy and geochemistry. Improved knowledge on the geochemical heterogeneity of low-permeable rocks is necessary to enhance the reliability of geological models used for assessing unconventional resources and designing effective production strategies [1].

The organic matter content is commonly assessed by combustion methods, such as total organic carbon (TOC) analysis and Rock-Eval pyrolysis [2]. Many techniques have been applied to provide qualitative and semi-quantitative analysis of kerogen chemical structure, such as vitrinite reflectance [3,4], nuclear magnetic resonance (SS NMR) [5,6] and combinations of flash pyrolysis with gas chromatography [7]. Vitrinite reflectance measurements require considerable time and expertise; moreover, it cannot be applied for marine shales, devoid of terrestrial organic matter input, and pre-Silurian shales, which are lacking in plant material [7]. High-field NMR spectroscopy analysis delivers insights into the kerogen structure, and the sample preparation, experiments and data processing are complex. Rock-Eval pyrolysis provides qualitative and quantitative data in which $T_{max}$ is the main maturity index. However, the technique has low sensitivity for highly mature and post-mature samples.

Transmission Fourier transform infrared (FTIR) spectroscopy has been used for many years from the middle of the last century to assess mineralogy [8] and the composition of organic matter in kerogen [9–15]. FTIR studies assign specific absorption bands to chemical structure fragments in complex kerogen molecules [16,17] and proved the diagnostic value for maturity, kerogen type, and oil/gas generation potential [10,11,14,18].

FTIR absorbance signals in kerogen and macerals were evaluated as indices for thermal maturity [19,20]. The correlation was observed between the aromatic/aliphatic absorption ratio and vitrinite reflectance $R_0$. FTIR parameters are especially valuable for determining the degree of maturity of marine source rocks lacking vitrinite. With the increasing maturity of organic matter, FTIR spectra express four trends: an increase in the absorption of aromatic bands, a decrease in the absorption of aliphatic bands, a loss of oxygenated groups (carbonyl and carboxyl), and an initial decrease in the $CH_2/CH_3$ ratio that is not apparent at higher maturity in naturally matured samples but is observed throughout increasing $R_0$ in artificially matured samples [19]. The $CH_2/CH_3$ index has good sensitivity from low maturity, the oil window and up to the condensate window. The technique does not require separating organic matter or identifying macerals, so it is more reliable than % $R_o$ [7]. For samples with different maturity, changes in spectra in some specific functional groups were observed: during maturation, organic matter converts into more aromatics and less aliphatics [7], along with the presence of a shorter aliphatic chain length [20]. The diffuse reflectance infrared Fourier transform spectroscopy (DRIFT) technique was implemented [21,22], because in the case of the DRIFT technique, the high frequency bands (4000–2400 $cm^{-1}$) are usually informative with respect to the detection of aromatic and aliphatic bands in the examined rocks. The occurrence of aromatic groups in the samples is the evidence of higher kerogen maturity [21]. Characteristic DRIFT signals for kerogen type II correspond to the bands centered at 3430, 3050, 2953, 2923, 2856, 1600, 1700, 1453, and 1375 $cm^{-1}$ frequencies [22]. Qualitative interpretation of the spectra was consistent with the known maturity and organic facies of the samples. For example, aliphatic signals (2953, 2923, 2856 $cm^{-1}$) were more intense in the less mature samples, whereas the aromatic signals (1604 $cm^{-1}$) were prominent in over-mature samples. Additionally, the carbonyl signal (1700 $cm^{-1}$) was much more pronounced in the samples, which were known to correspond to a region where redox conditions of the depositional environment were more oxic. The qualitative interpretation was confirmed by different compositional indices. Organic facies indices were shown by Chen et al. [14] to be influenced by maturity effects. The higher thermal maturity of samples was supported by increasing aromaticity (indices by Lis et al. [19]), as well as decreasing the content of aliphatic bonds. Uncertainty of the indices generally resulted in <5% variations [22].

The study of organic solid materials in geological systems (i.e., kerogen, bitumen, etc.) is an area of extensive research [23–26], but currently only a limited number of studies on the application of infrared spectroscopy to characterize organic and carbonaceous materials dispersed in shales have been published. Recent work has been devoted to direct determination of the organic matter in rock samples with minimal sample preparation [27–29]. However, very few of them have reported the appropriate procedure for the study of oil shales. The IR spectroscopic method was applied [30] for the determination of the organic geochemistry of shale rocks that contain low amounts of carbonaceous materials. Using linear regression analysis, the analytical performance of both ATR and DRIFT methods were almost comparable. DRIFT was found to be more sensitive for determining the organic carbon content in powdered rock samples but was prone to interferences and spectral overlap from the carbonate overtone band when compared to ATR. It was shown for shales containing low TOCs [30] that the carbonate-adjusted C-H absorption band intensity between 2800 and 3000 $cm^{-1}$ may be directly related to the geochemical parameters $S_1$ and $S_2$.

It is essential to use several complementary methods to reduce the uncertainty of the oil shales evaluation [7,21]. For example, thermal methods, including thermogravimetric analysis (TGA) and differential scanning calorimetry (DSC), made it possible to trace the

reactions during heating the samples in an oxidizing and inert atmosphere. Based on the differential thermogravimetry (DTG) curves, the maximum temperature of the pyrolysis peak could be determined, and as a consequence, the thermal maturity of the organic matter in the sample. The pyrolysis peak ($T_{max}$), which reflects the thermal maturity of the kerogen, was between 477 and 544 °C. The high $T_{ex}$ of pyrolysis in measurements of fine-grained Palaeozoic rocks was consistent with the FTIR results, which indicate the presence of aromatic hydrocarbons.

Recently, FTIR microscopy (micro-FTIR) was proposed as a rapid, non-destructive, and spatially resolved method for determining the mineralogy and organic matter content within shale samples [31]. Mineralogy and TOC could be predicted to within margins of error similar to those of the current accepted industry standard techniques and might be useful for the development of models that can be applied to well log data on mineralogy to provide an estimate of TOC along the borehole [31]. Mapped distributions of organic matter provide information on the organic matter abundance and the connectivity of organic matter within the overall shale matrix [1]. In addition, micro-FTIR, in combination with complementary porosimetric techniques, strengthens the understanding of porosity networks. Image analysis can be employed for fast and convenient characterization of organic matter in highly mature shales where the application of micro-FTIR is hampered by low aliphatic infrared absorption [1].

The aim of this work was to develop a non-destructive method for mapped distributive assessment of organic matter structure and content for shale samples using FTIR spectroscopy and FTIR microscopy. In this work, we have used both the transmission and the ATR modes for study by FTIR spectroscopy. It was important to test the feasibility of using ATR for organic matter content evaluation compared to transmission mode used for quantitative measurements. We have used FTIR transmission spectra to estimate the content according to the basic law of light absorption by the internal standard and the normalization methods. In addition, very few recent works have reported the appropriate procedure for organic matter study in rock samples. We have developed a procedure to assess the content and distribution of organic matter directly in the rocks through the analysis of vibration bands of C-C and C-H bonds of aliphatic and aromatic fragments in the organic matter of rock samples. The procedure has been applied for the study of organic matter in core samples of the Bazhenov Formation (West Siberia, Russia). It was shown that the aliphatic C-H stretching region (2950–2850 cm$^{-1}$), aromatic C-C stretching (~1600 cm$^{-1}$) or aromatic C-H deformation (876 cm$^{-1}$) region could be used for the evaluation of the organic matter maturity in rock and kerogen by transmission FTIR spectroscopy (in KBr pellets). Based on the obtained FTIR spectroscopy results, the possibility of using ATR FTIR microscopy for a semi-quantitative assessment of the organic matter content in rock without preliminary sample preparation was shown. The ATR FTIR microscopy results on polished shale samples also provide an insight into the heterogeneity of the organic matter distribution on the surface of rock, and aliphatic and aromatic fragment ratios. In general, it was demonstrated that FTIR spectroscopy and FTIR microscopy allow one to evaluate the organic matter content using the intensity of major aliphatic and aromatic C-H and C-C bands, which is agreed with pyrolysis and TGA data from the homogenized samples.

## 2. Materials and Methods

### 2.1. Source Rock and Kerogen

The objects of this study were clay-siliceous shale samples of the Bazhenov Formation of the West Siberian petroleum Basin (the Krasnoleninskiy Arch) from depths between 2710 and 2730 m and extracted kerogen (type-II, oil window) samples (mature, M). For comparative measurements, we used immature kerogen (type-II) sample (IM) from one field of the Nyurolskaya Depression from a depth of 2600 m. The Upper Jurassic–Lower Cretaceous Bazhenov Formation is regarded as the principal source of the West Siberian oil fields. Its age has been determined mostly as Tithonian (Volgian) to Lower Berriassian [32,33]. The

formation is spread over more than $10^6$ km$^2$ in the West Siberian Plate and its thickness ranges from 15 to 20 up to 50 to 60 m [34].

The shale samples were crushed to lower than 200 mesh size and extracted in the Soxhlet apparatus (Moscow, Russia) using chloroform ($CHCl_3$) for approximately 300 h. To remove carbonate minerals, grounded shale samples were treated with 36% HCl, heated in a water bath, washed with deionized water and dried at 95 °C. The extracted and decarbonated samples of 1 to 2 mg were subjected to organic geochemical analyses, namely Rock-Eval pyrolysis, elemental (CHNS), thermal analysis (TGA), bitumen extraction, and FTIR-spectroscopy in transmission mode.

### 2.2. Kerogen Preparation

Crushed source rock samples were extracted repeatedly with $CHCl_3$ to remove soluble OM (until the supernatant was no longer colored). The dry, extracted rock powder was demineralized using the HF-HCl method [35]. The technique is based on acid treatment of the rock with 10% HCl and concentrated hydrofluoric acid to remove carbonate, silicate and clay minerals and elemental sulfur. To remove pyrite, sulfide reduction was carried out using 10% chromium (II) chloride. The evolved hydrogen sulfide was absorbed with a cadmium acetate solution. Pyrite is the principal mineral that cannot be completely eliminated by this acid treatment procedure [36].

The purity of the kerogen concentrates was confirmed by X-ray diffraction analysis (XRD) and X-ray fluorescence spectroscopy (XRF), since the presence of pyrite in the kerogen concentrate affects the absolute geochemical composition of the bulk concentrate. However, the presence of pyrite does not affect the measured IR spectrum of kerogen because the IR absorbance bands of pyrite are small and do not overlap with those in the spectra of organic materials.

The isolated kerogen fractions were retained for subsequent geochemical and spectroscopic studies.

### 2.3. Fourier Transform Infrared Spectroscopy

Samples for FTIR were prepared as potassium bromide (KBr) pellets, containing 0.1 to 0.8 wt.% of powdered rock or kerogen. Potassium bromide (KBr) was grounded in agate mortar and kept in an oven at 650 °C for 6 to 8 h in order to remove water and other potential impurities. Mixtures of sample/KBr were prepared in a clean and dry agate mortar. The pellets were dried under a vacuum before the experiment. The analysis was performed on a Perkin Elmer Spectrum 100 spectrometer equipped with a diamond ATR accessory (Surgut, Russia) and a Shimadzu IRAffinity-1S spectrometer (Surgut, Russia). During acquisition, 25 scans per sample were collected at a resolution of 1 cm$^{-1}$.

Bands of absorption were identified by comparison with published spectra [1,16,17,36–38]. The following OM absorption bands (optical density) were measured: aliphatic C-H stretching region (3000–2800 cm$^{-1}$), the aromatic ring stretching (~1600 cm$^{-1}$), $CH_2$ and $CH_3$ bending modes (peak at 1450 cm$^{-1}$), $CH_3$ absorption bands (peak at 1375 cm$^{-1}$), and the aromatic out-of-plane C-H deformation region (700–900 cm$^{-1}$).

In order to define labels assigned to specific bands or band ratios, we use abbreviations. Abbreviations $AR_{1600}$ and $AR_{700–900}$ are assigned to aromatic regions, the aromatic ring stretching (1600 cm$^{-1}$) and aromatic C-H out-of-plane deformation bands (700–900 cm$^{-1}$), respectively. Abbreviations $AL_{2800–3000}$, $AL_{1450}$, and $AL_{1375}$ are assigned to the aliphatic C-H stretching region (2800–3000 cm$^{-1}$), $CH_2$ and $CH_3$ bending modes (1450 cm$^{-1}$), and $CH_3$ absorption (1375 cm$^{-1}$), respectively. The absorption band by 1700 cm$^{-1}$ is subsequently referred to as C=O stretch vibration in carbonyl/carboxyl groups.

The aromatic out-of-plane C-H deformation region (700–900 cm$^{-1}$) in shale and some kerogen samples, containing residual mineral components after demineralization by the commonly used HCl-HF method, were overlapped by strong absorption bands of quartz, carbonate (calcite, dolomite) and clay minerals.

FTIR spectra were baseline-corrected and intensity-normalized for organic matter content evaluation in kerogen and shale samples using aliphatic C-H stretch (2923–2926 cm$^{-1}$), aromatic C-H deformation (876 cm$^{-1}$) or C-C ring stretch (1630 cm$^{-1}$), Si-O of clay (1100–1200 cm$^{-1}$) and Si-O and Si-O-Si of silicate (430–800 cm$^{-1}$) minerals. In the normalization method, the content of organic and mineral constituents (wt.%) was calculated as the ratio of corresponding peak intensity to the sum of the major IR band intensities. The organic matter content was evaluated as the sum of aliphatic and aromatic fragment contents.

The internal standard method with KSCN (0.5 wt.%) was also used for the evaluation of organic matter content in kerogen samples. KSCN was chosen as standard as its strong stretch bands did not overlap aliphatic and aromatic band regions. Mixtures of sample/KBr/KSCN (0.4 wt.% kerogen and 0.5 wt.% KSCN) were grounded in agate mortar and pressed in pellets under a vacuum. A strong C-N stretch band at 2060 cm$^{-1}$ was used as the standard peak. The content (wt.%) of aliphatic or aromatic fragments was calculated by multiplying the ratio of the peak intensity of corresponding C-H stretch (2923–2926 cm$^{-1}$) or C-H deformation (876 cm$^{-1}$) bands to the intensity of the standard peak by the content of the standard (wt.%) in the KBr pellet. When comparing the results for several samples, using the relative intensity of the absorption bands, the ratio of the absorption coefficients for the bands of functional groups was neglected.

The results of normalization and internal standard methods were compared with TOC and thermal analysis data. In this study, TGA was executed using two heating programs: standard TGA program (the temperature range of 20 to 1000 °C; a heating rate of 10 °C/min in the oxidizing atmosphere (synthetic air) or a heating rate of 25 °C/min in the inert gas (nitrogen) with a gas flow rate of 50 mL/min), and the same standard thermogram of Rock-Eval$^{TM}$ (heating to 300 °C under inert gas (nitrogen), and holding there for 5 min; followed by heating at a rate of 25 °C/min until a temperature of about 650 °C, holding there for 5 min and cooling up to 100 °C; followed by heating at a rate of 25 °C/min up to 850 °C in the oxidizing atmosphere). The samples were powdered and put into the alumina (Al$_2$O$_3$) crucible in amount of about 20 mg.

### 2.4. Fourier Transform Infrared Microscopy

Measurements were performed using a Shimadzu AIM-9000 FTIR microscope (Surgut, Russia) in the ATR mode (Shimadzu Corporation, ATR objective Ge prism). The spot size for all measurements was 100 × 100 μm. The spectral window ranged from 700 to 4000 cm$^{-1}$, with a resolution of 4 cm$^{-1}$. FTIR spectra were baseline-corrected and the absorption band intensity was normalized (IRSolution, Shimadzu) before use in model development or property prediction; no other preprocessing was performed.

A polished sample of the Bazhenov Formation was used for the ATR measurements by FTIR microscopy (micro-FTIR). A total of 110 spectral measurements were collected, spaced 100 μm across the laminae of the Bazhenov Formation sample.

Predicted organic matter content values based on infrared spectra used spectral regions associated with aliphatic and aromatic carbon C-H and C-C stretches. These values were compared with TOC and simultaneous thermal analysis data.

### 2.5. Methods for Geochemical Characterization of Samples

Geochemical characteristics of kerogen and shale samples were performed using TOC content, Rock-Eval pyrolysis, elemental (CHNS) and thermal analyses.

The total organic carbon (TOC, wt.%) content, T$_{max}$ (°C), S$_1$ and S$_2$ (mg HC/g rock) were measured using a Rock-Eval pyrolysis instrument (HAWK Resource Workstation, Moscow, Russia) according to the procedure of Espitalié et al. [39]. The parameters S$_1$ (number of free hydrocarbons), S$_2$ (number of hydrocarbons generated by thermal cracking of non-volatile organics), T$_{max}$ (temperature of maximum hydrocarbon release during kerogen cracking), and TOC were measured during pyrolysis. The production index (PI = S$_1$/S$_1$ + S$_2$) was calculated according to [40].

The H/C ratios of the kerogen samples were determined using a Perkin Elmer 2400 Series II CHNS/O Analyzer (Surgut, Russia). The fine-grounded samples were analyzed using an elemental analyzer to measure CHNS contents in weight percentage. The atomic ratios of H/C were then calculated following the procedure from ASTM D5373-08 [41].

Simultaneous thermal analysis (STA) is a simultaneous implementation of two thermal analysis techniques: differential scanning calorimetry (DSC) and thermogravimetry analysis (TGA). A TGA/DSC 3+ Star System (Mettler Toledo) setup was used to determine mass changes and heat effects during pyrolysis and oxidation.

## 3. Results and Discussion

### 3.1. IR Spectra of Kerogen and Shale Rock Samples

Five kerogen samples and 6 shale samples from the Bazhenov Formation were studied by FTIR spectroscopy in transmission and ATR modes (Figures 1a,b and 2a,b). Bulk characteristics of organic matter from these samples are shown in Tables 1 and 2.

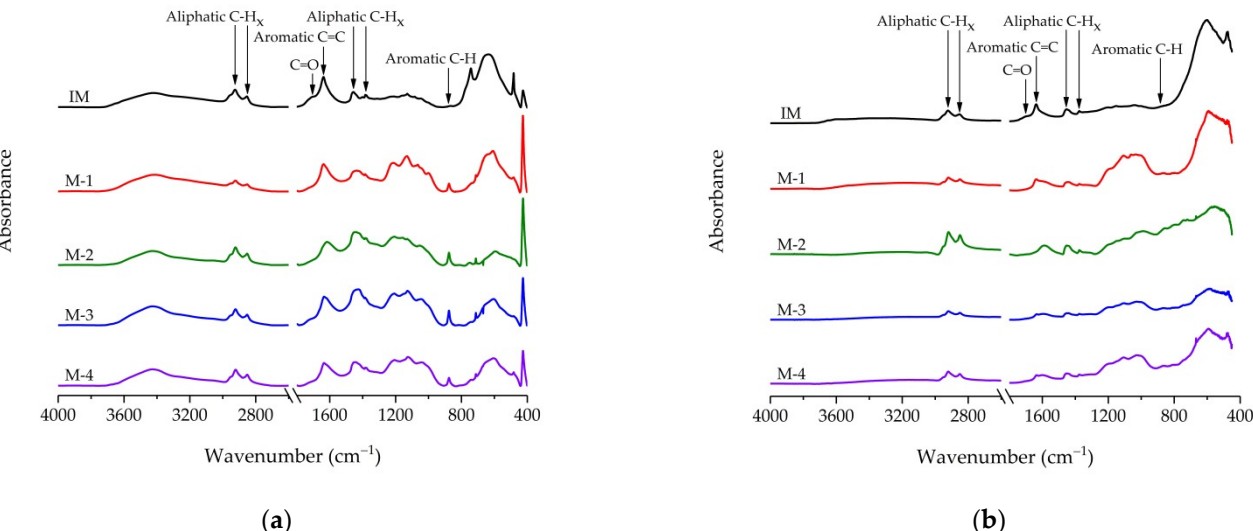

**Figure 1.** FTIR spectra of the Bazhenov formation kerogen samples from Krasnoleninskiy Arch (M-1–M-4) and Nyurolskaya Depression (IM): (**a**) in transmission mode (0.1% kerogen at KBr pellet); (**b**) in ATR mode.

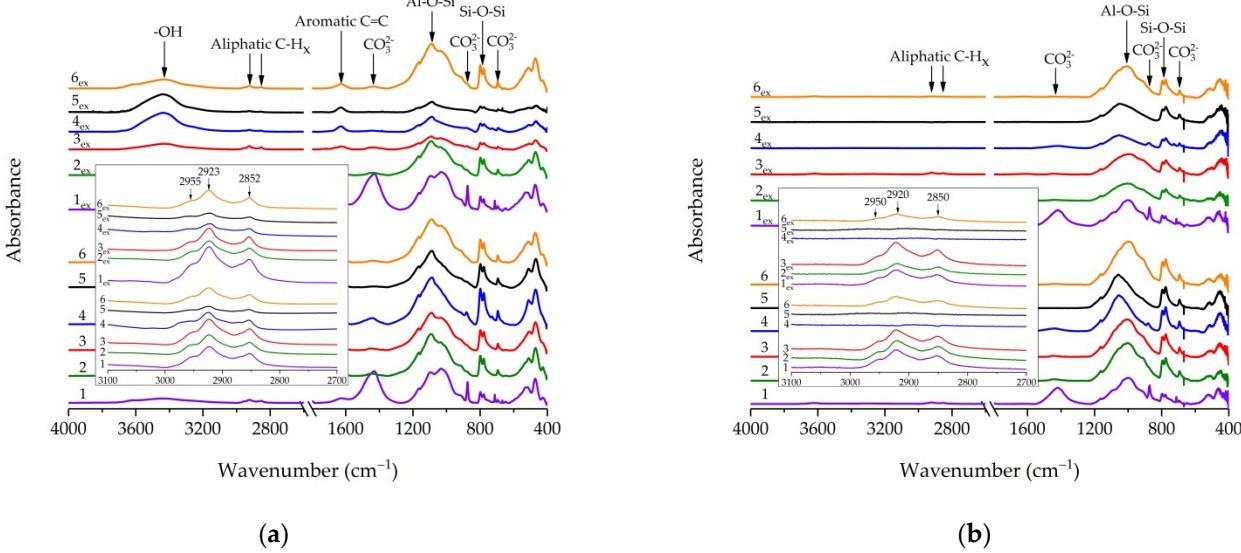

**Figure 2.** FTIR spectra of the Bazhenov Formation rock samples before (1–6) and after ($1_{ex}$–$6_{ex}$) bitumen extraction: (**a**) in Table 3. sample at KBr pellet; (**b**) in ATR mode.

**Table 1.** General characteristics of kerogen samples.

| Sample | Depth, m | TOC, wt.% | $T_{max}$, °C | $S_1$, mgHC/g Rock | $S_2$, mgHC/g Rock | H/C |
|--------|----------|-----------|---------------|--------------------|--------------------|-----|
| IM | 2600 | 35.20 | 421 | 2.89 | 189.11 | 1.29 |
| M-1 | 2721 | 34.51 | 440 | 1.34 | 59.02 | 1.19 |
| M-2 | 2725 | 38.46 | 444 | 1.03 | 94.06 | 0.92 |
| M-3 | 2730 | 35.22 | 442 | 2.38 | 55.86 | 0.93 |
| M-4 | 2732 | 27.60 | 439 | 0.71 | 51.92 | 1.02 |

**Table 2.** General characteristics of shale rock samples before and after bitumen extraction. Abbreviations $1_{ex}$–$6_{ex}$ are assigned to extracted rock samples 1–6, respectively.

| Sample | Depth, m | TOC, wt.% | $T_{max}$, °C | $S_1$, mgHC/g Rock | $S_2$, mgHC/g Rock | PI | H/C |
|--------|----------|-----------|---------------|--------------------|--------------------|-----|-----|
| 1 | 2717 | 13.43 | 447 | 4.27 | 52.04 | 0.08 | 0.92 |
| $1_{ex}$ | | 12.43 | 446 | 0.14 | 40.48 | - | 0.91 |
| 2 | 2719 | 11.62 | 443 | 4. 41 | 37.95 | 0.10 | 1.18 |
| $2_{ex}$ | | 9.58 | 445 | 0.26 | 27.22 | - | 1.17 |
| 3 | 2723 | 12.25 | 446 | 2.48 | 44.67 | 0.05 | 1.18 |
| $3_{ex}$ | | 8.20 | 446 | 0.12 | 43.61 | - | 1.20 |
| 4 | 2725 | 2.87 | 441 | 3.11 | 4.18 | 0.43 | 0.85 |
| $4_{ex}$ | | 1.97 | 446 | 0.06 | 2.08 | - | 0.71 |
| 5 | 2726 | 2.44 | 446 | 2.20 | 3.47 | 0.39 | 1.58 |
| $5_{ex}$ | | 1.91 | 452 | 0.11 | 2.24 | - | 1.51 |
| 6 | 2727 | 6.58 | 445 | 1. 49 | 15.38 | 0.09 | 1.32 |
| $6_{ex}$ | | 6.60 | 442 | 0.11 | 17.21 | - | 1.30 |

For the isolation of kerogen, we treated the whole rock samples with concentrated acids to remove inorganic mineral components. XRD and XRF data indicated the presence of residual pyrite (up to 5.9 wt.%), clay and siliceous minerals in kerogen, which are common in concentrated kerogen isolated according to the given procedure [35,36]. One kerogen sample from the Nyurolskaya Depression and four kerogen samples from the Krasnoleninskiy Arch have been examined (Table 1).

The atomic H/C ratios in kerogen samples determined by elemental analysis are given in Table 1. The H/C ratio for kerogen is unaffected by the presence of mineral admixture. It can be noted that the H/C ratio of immature kerogen from the Nyurolskaya Depression (sample IM) is 1.29, which is slightly higher than that of mature kerogen from the Krasnoleninskiy Arch ~1.00 (samples M-1–M-4).

The absorbance scale for IM and M-1–M-4 kerogen samples in transmission mode (Figure 1a) is absolute and the same. We have found several absorption bands of functional groups for kerogen samples, reflecting the gross structure of kerogen. The IR absorbance band at 1700 cm$^{-1}$ is attributed to the presence of oxygen-containing bonds C=O; 2925, 2852, 1457 and 1376 cm$^{-1}$ are related to the stretching and bending bands of aliphatic C-H$_x$ groups; 1600−1630 and 700−900 cm$^{-1}$ are related to aromatic C=C ring stretching and C-H bending signals.

The asymmetric and symmetric methylene (CH$_2$) stretching (2925 and 2852 cm$^{-1}$), aliphatic C-H bending (1457 cm$^{-1}$) and aromatic C=C ring stretching (1630 cm$^{-1}$) bands are more pronounced in IR spectra of kerogen samples in ATR mode compared to transmission mode (Figure 1b). However, the band intensity for the spectra collected in ATR mode is noticeably lower. We have noticed that there are no aromatic C-H bending signals in ATR spectrum of immature kerogen (IM) in comparison with M kerogen samples, which is attributed to the low absorption coefficient of the aromatic C-H deformation band resulting in the low sensitivity of the method.

The 6 selected rock samples were studied both before and after bitumen extraction (Table 2). Abbreviations $1_{ex}$–$6_{ex}$ are assigned to extracted rock samples 1–6, respectively.

The selected shale rock samples for the study are characterized by TOC content from 2 to 13 wt.% (Table 2). The Bazhenov section is represented by rocks with a high organic matter content, which is classified as type II kerogen with excellent oil-generating properties. The values of $T_{max}$ are from 440 to 446 °C. The degree of organic matter maturity corresponds to the maximum oil generation.

Shale rocks are heterogeneous and have a complex multicomponent composition. Figure 2 shows IR spectra of the studied samples obtained using transmission (a) and ATR (b) spectroscopy. The majority of the IR bands that appear in the mid-infrared region are attributed to mineral/inorganic components (between 4000 to 400 cm$^{-1}$). The absorption bands of transmission spectra at 1431, 876, and 712 cm$^{-1}$ predominantly correspond to the asymmetric stretching mode of the $CO_3^{2-}$ ion, out-of-plane bending vibration and in-plane bending vibration, respectively, and this confirms the presence of carbonate species in the samples (calcite, dolomite, etc.) (Figure 2a). An intense band at 1200–900 cm$^{-1}$ has been detected in all samples, which is characteristic for shale rocks and is assigned to the Si-O bond arising from the silicate ($SiO_4^{2-}$). The peak is generally broad and contains a number of different vibration modes due to silicates (Si-O-Si stretch) and clays (Si-O-Al stretch). The peaks at 798, 779, and 695 cm$^{-1}$ are attributed to the various Si-O-Si vibrations that are typical of silicates such as quartz [42,43]. The peaks from 600 to 400 cm$^{-1}$ are broadly characterized as O-Si-O bending vibrations [43]. In ATR spectra of shale samples (Figure 2b), the same bands of mineral components can be observed, but their intensity is significantly lower. Therefore, ATR is more convenient for direct probing of the samples without sample preparation compared to the traditional transmission method, which requires the preparation of KBr pellets. Thus, ATR can be particularly useful for the characterization of the complex multicomponent composition of rocks, for both organic and mineral components.

A number of weak bands attributed to organic components in the shale rock were observed from 3000 to 2800 cm$^{-1}$ and from 1600 to 1630 cm$^{-1}$ in transmission spectra (Figure 2a). The asymmetric methyl (CH$_3$) stretch gives rise to bands at ~2955 cm$^{-1}$. Asymmetric and symmetric methylene (CH$_2$) stretches occur at ~2923 and ~2852 cm$^{-1}$, respectively. The aromatic C=C ring stretching vibrations appear around 1600 cm$^{-1}$. It is worth noting that the aliphatic C-H stretching bands are very weak in ATR spectra (Figure 2b) and almost non-visible in the samples with TOC lower than 3 wt.%. Further examination of the ATR spectra in the region from 1800 to 1400 cm$^{-1}$ did not reveal any significant bands. This is consistent with organic matter being present at concentrations that are near to the limit of detection of the ATR technique.

Figure 2a,b shows insignificant differences in the band intensities and peak shapes in IR spectra of shale samples after bitumen extraction compared to spectra of the samples before it. After removal of carbonate minerals using 36% HCl, the lower carbonate band at 1431 cm$^{-1}$ can be observed in the transmission spectra (Figure 3). There is an absence of the asymmetric stretching band of the $CO_3^{2-}$ ion at 876 cm$^{-1}$ in IR spectra after acid treatment of shale. Organic matter C-H and C=C stretching and bending bands are clearly evident in the transmission spectra at 2922, 2854, 1624 and 1380 cm$^{-1}$ after removing the carbonate minerals. This approach expands the IR spectroscopy method for studying organic matter structure directly in rocks without the preliminary stage of kerogen isolation.

Thus, the comparison of the FTIR results of studied samples shows several absorption bands attributed to organic matter for both kerogen and shale samples: stretching and bending of aliphatic C-H$_x$ and stretching of aromatic C=C ring bonds. After removing the carbonate minerals by 36% HCl, the bands in transmission spectra became more resolved and could be used for organic matter structure investigation directly in the rock sample. It was shown that in ATR spectra of rock samples, C-H stretching bands are very weak and almost non-visible in samples with low TOC (<3 wt.%) compared to strong bands attributed to the mineral components.

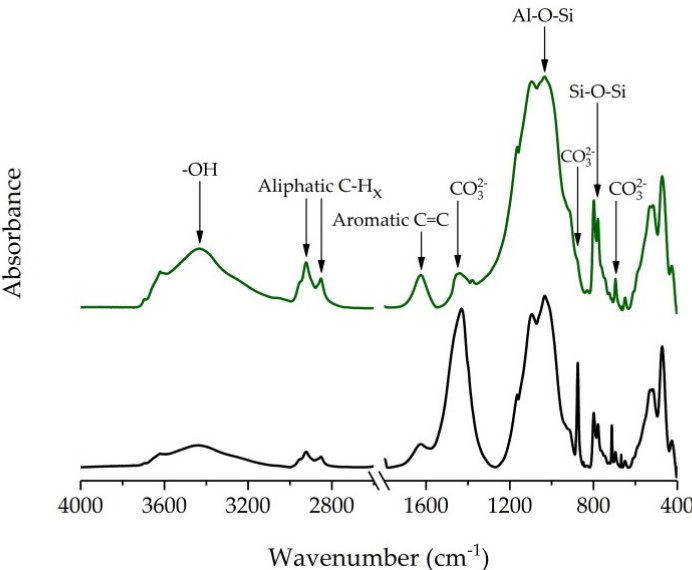

**Figure 3.** FTIR transmission spectra of the Bazhenov Formation shale rock (sample 1) before (black curve) and after removing carbonate minerals using 36% HCl (green curve).

### 3.2. Evaluation of Organic Matter Content by IR Spectroscopy

The kerogen (IM and M-1–M-4) and shale rock samples after acid treatment, before and after bitumen extraction (1–6 and $1_{ex}$–$6_{ex}$, respectively) were studied by FTIR spectroscopy in transmission mode. The direct evaluation of organic matter content was performed by Rock-Eval pyrolysis and TGA using the standard thermal program of Rock-Eval$^{TM}$ bulk rock (Program 1), two standard programs in oxidizing atmosphere (program 2) and in an inert gas (Program 3) (Figures 4–6, Tables 3 and 4 and Appendix A, Figures A1–A4). The results were compared to develop the procedure of TOC determination using FTIR spectroscopy.

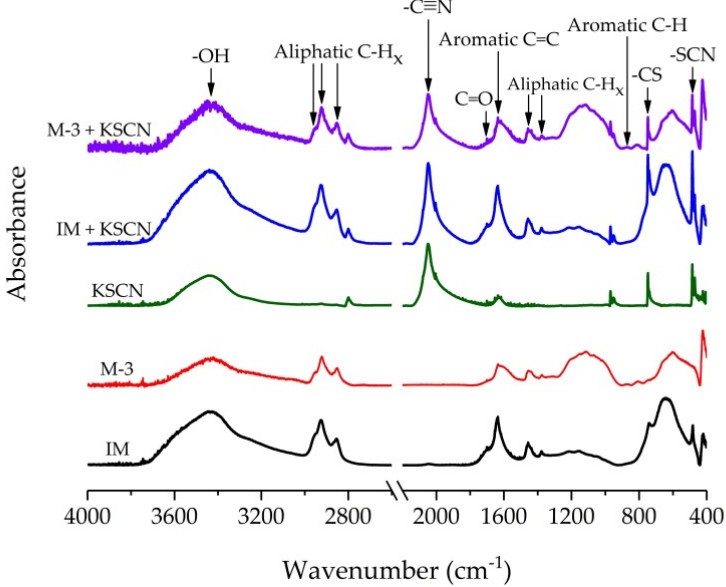

**Figure 4.** FTIR spectra of the Bazhenov Formation kerogen samples (IM and M-3) in transmission mode (0.8% sample in KBr pellet) in the presence of the internal standard KSCN (0.5% KSCN in KBr pellet).

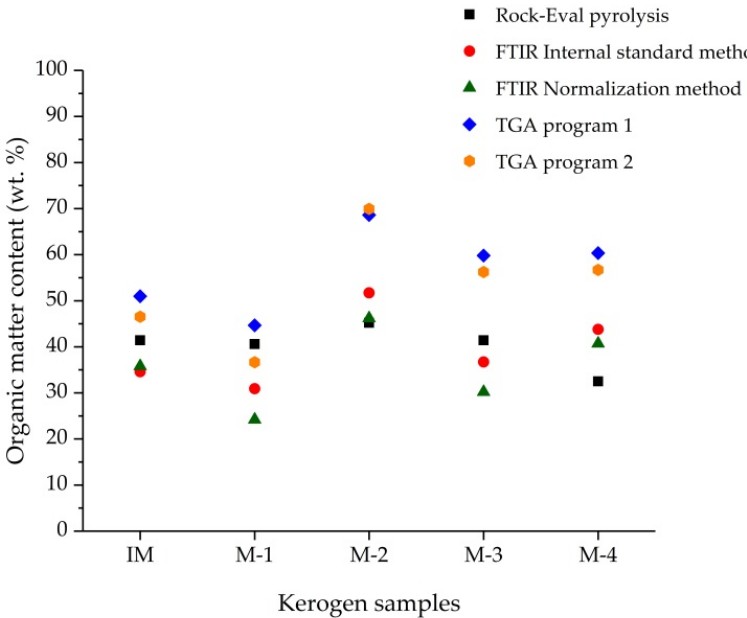

**Figure 5.** Comparison of organic matter (OM) content in kerogen samples from the Nyurolskaya Depression (IM) and the Krasnoleninskiy Arch (M-1–M-4) evaluated by Rock-Eval pyrolysis, TGA (Program 1: the same standard thermogram of Rock-Eval$^{TM}$ and Program 2: standard program in the oxidizing atmosphere) and FTIR spectroscopy in transmission mode (normalization and internal standard methods).

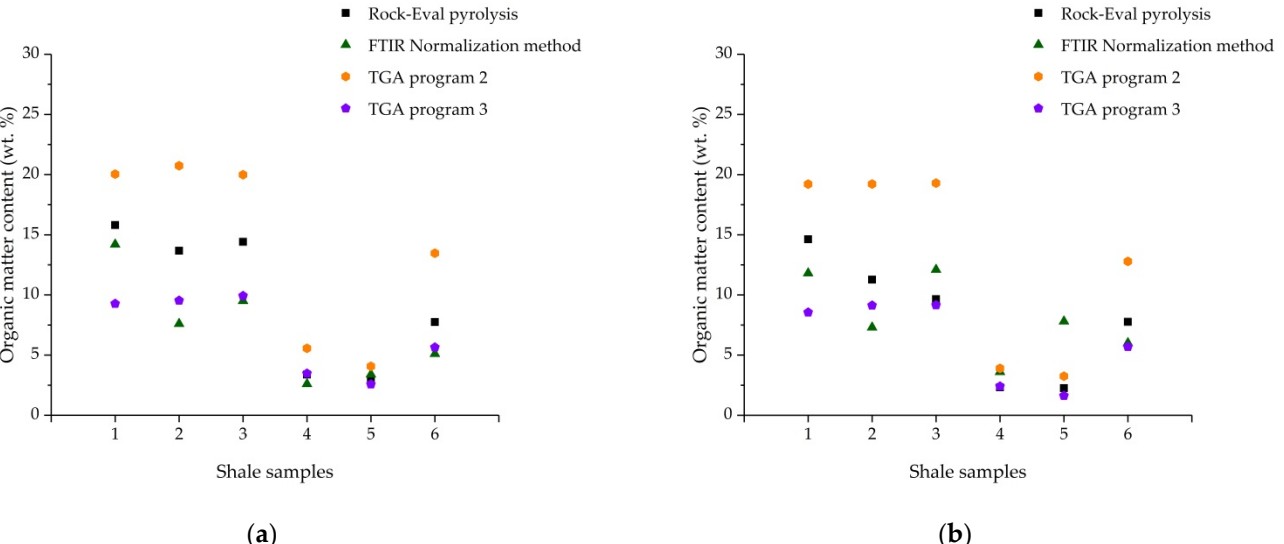

(**a**)                                                    (**b**)

**Figure 6.** Comparison of organic matter (OM) content in the Bazhenov Formation rock samples after acid treatment, 1–6 (**a**) before and $1_{ex}$–$6_{ex}$ (**b**) after bitumen extraction, evaluated by Rock-Eval pyrolysis, TGA (Program 2: in the oxidizing atmosphere; and Program 3: in the inert gas) and FTIR spectroscopy in transmission mode (normalization method).

The shape of the aliphatic band in the 3000–2800 cm$^{-1}$ region was different for some of the studied samples, possibly due to the presence of various types of organic matter in the shale samples. Figure 4 presents the IR spectra of two kerogen samples (Table 1) that contain the same total organic carbon content (35.20 and 35.22 wt.%, respectively) but show significant differences in maturity and C-H peak intensity. The intensity of the asymmetric CH$_2$ stretch (at ~2925 cm$^{-1}$) relative to the asymmetric CH$_3$ stretch (at ~2955 cm$^{-1}$) variations most probably result from different thermal maturity of kerogen/organic matter in these two samples. Previous studies have shown that both the C-H intensity and CH$_2$/CH$_3$

ratio decrease considerably with increasing vitrinite reflectance/maturity [27,44,45]. The region between 3000 and 2800 cm$^{-1}$ is mainly associated with C-H bonds and it does not contain all of the molecular vibrations arising from organic compounds (i.e., aromatic hydrocarbons), which may also be present in the kerogen sample and contribute to the TOC content. The aromatic band (876 cm$^{-1}$) in kerogen samples corresponds to the aromatic C-H deformation and is weak due to its low absorption coefficients.

**Table 3.** Evaluation of organic matter (OM) content and composition in kerogen samples by Rock-Eval pyrolysis, TGA and FTIR spectroscopy in transmission mode (0.8 wt.% sample in KBr pellet) in the presence of the internal standard KSCN (0.5 wt.% KSCN at KBr pellet).

| Sample | TOC by Rock-Eval, wt.% | OM Content Evaluated by FTIR, wt.% | | Mass Loss by TGA, wt.% (up to 650 °C) | |
|---|---|---|---|---|---|
| | | Internal Standard Method (KSCN) | Normalization Method | Program 1 (N$_2$, Air) | Program 2 (Air) |
| IM | 35.20 | 34.6 | 35.8 | 50.97 | 46.54 |
| M-1 | 34.51 | 30.9 * | 24.2 * | 44.66 | 36.66 |
| M-2 | 38.46 | 51.7 * | 46.2 * | 68.62 | 69.93 |
| M-3 | 35.22 | 36.7 | 30.2 | 59.82 | 56.25 |
| M-4 | 27.60 | 43.8 | 40.7 | 60.34 | 56.69 |

* 0.4% sample in KBr pellet.

**Table 4.** Comparison of aliphatic and aromatic fragments content and composition measurements in rock samples (after acid treatment) by Rock-Eval pyrolysis, TGA and FTIR spectroscopy in transmission mode (0.25 wt.% sample in KBr pellet).

| Sample | AL, AR and Minerals Content Evaluated by FTIR, wt.% | | | | | TOC, by Rock-Eval, wt.% | Mass Loss by TGA, wt.% (up to 650 °C) | |
|---|---|---|---|---|---|---|---|---|
| | Normalization Method | | | | | | Program 2 (air) | Program 3 (N$_2$) |
| | AL$_{2925}$ | AR$_{1620}$ | Clay | Quartz | Other Silicates | | | |
| 1 | 8.3 | 5.9 | 42.4 | 19.5 | 23.9 | 13.43 | 20.04 | 9.27 |
| 1$_{ex}$ | 6.5 | 5.3 | 42.3 | 22.5 | 23.5 | 12.43 | 19.21 | 8.55 |
| 2 | 3.3 | 4.3 | 43.6 | 20.4 | 28.4 | 11.62 | 20.74 | 9.54 |
| 2$_{ex}$ | 3.5 | 3.8 | 42.6 | 23.9 | 26.3 | 9.58 | 19.22 | 9.13 |
| 3 | 4.6 | 4.9 | 41.5 | 20.7 | 28.3 | 12.25 | 19.99 | 9.93 |
| 3$_{ex}$ | 5.1 | 7.0 | 41.1 | 19.6 | 27.2 | 8.20 | 19.29 | 9.15 |
| 4 | 1.2 | 1.4 | 41.4 | 25.7 | 30.2 | 2.87 | 5.56 | 3.47 |
| 4$_{ex}$ | 1.3 | 2.3 | 34.9 | 30.9 | 30.5 | 1.97 | 3.90 | 2.40 |
| 5 | 1.4 | 2.0 | 44.5 | 23.3 | 28.9 | 2.44 | 4.07 | 2.57 |
| 5$_{ex}$ | 2.7 | 5.1 | 39.7 | 24.9 | 27.6 | 1.91 | 3.24 | 1.62 |
| 6 | 2.8 | 2.3 | 39.9 | 24.3 | 30.7 | 6.58 | 13.47 | 5.65 |
| 6$_{ex}$ | 3.0 | 3.0 | 38.9 | 25.9 | 29.6 | 6.60 | 12.79 | 5.69 |

The aliphatic (2923–2926 cm$^{-1}$) and aromatic (1620–1630 cm$^{-1}$) bands in shale samples after acid treatment (Figure 3; see the example of sample 1) correspond to aliphatic C-H and aromatic C-C ring stretches, respectively. The weak aromatic C-H deformation band in FTIR spectra of the shale samples overlaps with broad and strong bands of silicates (Si-O-Si stretch) and clays (Si-O-Al stretch) in the rock.

All the kerogen samples were analyzed under oxidizing atmosphere (synthetic air) using standard TGA program (Table 3, Program 2) and temperature program as in Rock-Eval Bulk rock (Table 3, Program 1). Figure A1 (Appendix A) presents the example of TGA/DSC measurement for kerogen samples (IM and M-4): thermogravimetry (TG) and differential scanning calorimetry (DSC) curves. The set of green curves presents the measurement in oxidizing atmosphere (Table 3, Program 2). Two TG red curves are obtained during two-stage analysis that mimic Rock-Eval Bulk rock program: the first

stage is heating of the sample in inert gas (nitrogen) which is followed by the second stage with the oxidation of the residual organic carbon (Table 3, Program 1).

We observed the small mass loss of about 4 to 4.5% in the range from 20 to 200 °C, regardless of the experiment conditions. This effect is connected to water loss during the dehydration of clays and evaporation from rock pore space (weak endothermic peak 100–200 °C). The decomposition of kerogen starts in the range of 200 to 300 °C (Appendix A, Figure A1). The DSC green curves, which are obtained in course of the measurement under oxidizing atmosphere, have two intense exothermal peaks, indicating the combustion of organic matter at 240 °C and 290 °C. Over the temperature of 300 °C, the reactions of the organic matter combustion have been continued (exothermal effect on DSC curve at ~350 °C). In the case of the kerogen sample, this is a multi-step process (as a result of heterogeneity the organic matter), and is spanned between 200 and 650 °C. This effect is connected to mass loss (46.8–56.7%), which is visible on TG and DTG curves (Appendix A, Figure A1). Above the temperature of 650 °C, the further mass decline may be associated with the process of the dehydroxylation of clay mineral residuals in kerogen.

Under an inert atmosphere of the Rock-Eval program (red curves in Appendix A, Figure A1) in the range of 300 to 650 °C, the mass loss is around 20.3 to 26.0%, which is lower than that in the case of the oxidizing atmosphere experiment of TGA program. Three exothermal peaks (315, 360 and 525 °C) and the endothermic peak at 548 °C in DSC red curves are connected to the decomposition of organic matter in kerogen. Additionally, under an oxidizing atmosphere of the Rock-Eval program (green curves in Appendix A, Figure A1) in the range of 300 to 650 °C, the mass loss (20.9 to 37.8%) is lower than in the case of an inert atmosphere. The wide intense exothermal peak at 420 °C (from 360 to 540 °C, DSC green curve) is attributed to the combustion of residual kerogen.

In the case of shale samples (Appendix A, Figure A2), the intense exothermal effect is caused by the combustion of the organic matter under an oxidizing atmosphere (green curves) in TGA; it is a multi-step process between 300 and 600 °C. Under an inert atmosphere (red curves in Appendix A, Figure A2) in the range of 400 to 650 °C, the weak endothermic peak is attributed to the pyrolysis of organic matter. In the range of 650 to 850 °C, the further mass decline is associated with the loss of water associated with the dehydroxylation of clay minerals and decomposition of carbonates (Appendix A, Figure A2).

The obtained differential thermogravimetry (DTG) curves (Appendix A, Figure A3) reflect both the amount and type of organic matter, as well as clay and carbonate minerals content. The peaks within the range of 400 °C to 600 °C reflect the release of gaseous components from the sample during the pyrolysis process. In the case of rock sample 1 (Appendix A, Figure A3), the maximum yield of pyrolysis products is at 480 °C ($T_{max}$), which is evident on the DTG curve, and the $T_{max}$ of combustion is 423 °C (Appendix A, Figure A2). Skala et al. [46] found two peaks of endothermic heat absorption on DSC curves. The first was observed in the temperature range of 340 to 380 °C for kerogen type I-II, and at temperatures slightly higher than 400 °C, there was one for kerogen type II. The second was at temperatures above 450 °C. The second peak is the main endo-heat effect of pyrolysis. During our study, the first peaks were not detected. The second peak is not pronounced on DSC curves, but is clearly visible on DTG curves (480 to 490 °C), as a conspicuous mass loss. Based on the obtained results, the thermal maturity and type of organic matter could be determined, and the analyzed oil shale rocks contain kerogen type II.

Aliphatic (AL) and aromatic (AR) content evaluation by FTIR spectroscopy (normalization and internal standard methods) was carried out using the absorption intensity (optical density value) of aliphatic and aromatic bands in IR spectra. For kerogen samples, by the normalization method, we have used the major IR bands (Figure 4): aliphatic C-H stretch (2923–2926 cm$^{-1}$), aromatic C-H deformation (876 cm$^{-1}$), clay (~1130 cm$^{-1}$) and silicate (~600 cm$^{-1}$) residual minerals. Moreover, we have evaluated the content (in wt.%) of aliphatic and aromatic fragments in kerogen by the internal standard method using KSCN,

using relative intensities of aliphatic C-H stretch and aromatic C-H deformation bands at 2923 to 2926 cm$^{-1}$ and 876 cm$^{-1}$, respectively. The aliphatic and aromatic fragments content of organic matter in rock samples after acid treatment have been evaluated by the normalization method using the major IR bands (Figure 3, Appendix A, Figure A4): aliphatic C-H stretch (2923–2926 cm$^{-1}$), aromatic C-C ring stretch (1620–1630 cm$^{-1}$), clay (1020–1100 cm$^{-1}$), quartz (~695 cm$^{-1}$) and other silicate (~470 cm$^{-1}$) minerals. The AL:AR ratio was calculated as the ratio of aliphatic and aromatic contents and the organic matter content was calculated as their sum (Figures 5 and 6).

The mass loss in the temperature interval from 200 °C to 650 °C by TGA and TOC/0.85 value by pyrolysis were used for evaluating of organic matter content in kerogen (Figure 5) and shale rock samples (Figure 6).

The organic matter content evaluated by FTIR using the normalization method in kerogen samples is from 24% (M-1) to 46% (M-2) (Table 3). The FTIR results obtained by the internal standard method (from 31% to 52%) are in good agreement with those of the normalization method, as well as Rock-Eval pyrolysis (Figure 5). It should be noted that the organic matter content of some kerogen samples (IM, M-1 and M-3) by FTIR are slightly lower compared to that by Rock-Eval pyrolysis (~41%). We believe that FTIR gives underestimated results due to the low absorption coefficient of aromatic C-H deformation (876 cm$^{-1}$), which can lead to underestimated values of the aromatic fragments content in studied kerogen samples (Table 3, Figure 5). The results of TGA (particularly using program 1 in inert and oxidizing atmosphere) are significantly higher compared to FTIR and TOC (Table 3, Figure 5), which is possibly associated with thermal destruction and oxidative degradation of sulfur-containing organic substances in studied kerogen samples (21 ± 4 wt.% of S).

The aliphatic and aromatic fragments content evaluated by FTIR (normalization method) in the studied rock samples (after acid treatment) is from 1 to 8 wt.% and from 1 to 6 wt.%, respectively (Table 4). The organic matter content calculated as the sum of aliphatic and aromatic fragments values by FTIR are in good agreement with that by Rock-Eval pyrolysis and TGA (Program 3, inert gas) results (absolute measurement error does not exceed 6%, Table 4 and Figure 6a,b). As can be seen, the use of the aromatic C-C ring stretching (1620–1630 cm$^{-1}$) band to evaluate the organic matter content in rock samples, both before (1–6, Figure 6a) and after (1$_{ex}$–6$_{ex}$, Figure 6b) bitumen extraction, leads to good agreement with the results by Rock-Eval pyrolysis and TOC (Table 4). However, the error in this case may be due to overlapping with the deformation band of water (1620–1640 cm$^{-1}$). The TGA (Program 2, oxidizing atmosphere) results were 1.5 to 2.4 times higher compared to TGA (Program 3, inert gas) and Rock-Eval pyrolysis (or TOC) results due to the oxidation of pyrite and sulfur-containing organic compounds in oil shale samples.

The aromatic fragments content in some rock samples (after bitumen extraction and acid treatment) has significantly increased (Appendix A, Figure A4), which led to a decrease of the AL:AR ratio in the organic matter of the studied samples from 0.92 ± 0.29 to 0.76 ± 0.28. For instance, an AL:AR ratio of 3, 4 and 5 rock samples decreased 1.3 to 1.5 times (Table 4). Therefore, it could be applied for studying of the kerogen chemical structure and maturity of organic matter directly in the rock. It should be noted that FTIR spectroscopy using the normalization method also makes it possible to evaluate the content of minerals in shale samples: 41.0 ± 1.6% clay, 23.5 ± 2.1% quartz and 27.9 ± 1.5% other silicate minerals. Another advantage of FTIR spectroscopy for the source rock analysis is that it does not require the complete destruction of the sample chemical structure compared to the bulk methods such as pyrolysis and TGA, and the prepared KBr pellets can be reused.

The disadvantages of this approach include the probability of a negative error in the evaluation of aromatic fragments content using the C-H deformation band in kerogen, the low sensitivity of the ATR method for the evaluation of organic matter in the samples containing less than 3 wt.% TOC, and the requirement of acid pre-treatment for FTIR microscopy evaluation of organic matter and minerals content after the removal of carbonates.

### 3.3. Surface Organic Matter Analysis of Oil Shale Samples by FTIR Microscopy

To study the possibilities of the rock surface analysis by FTIR microscopy (micro-FTIR), we have used a polished layered sample from the Bazhenov Formation with $9 \times 13$ mm size (Figure 7). Alternating dark- and light-colored rocks can be clearly seen with the unaided eye in the sample laminae, and they are typically ~1.0 to 3.5 mm thick. The ATR spectra by FTIR microscopy of this sample (Figure 8) have been recorded in 25 dots along the length in dark (11 measurements) and light (14 measurements) laminae (Figure 7).

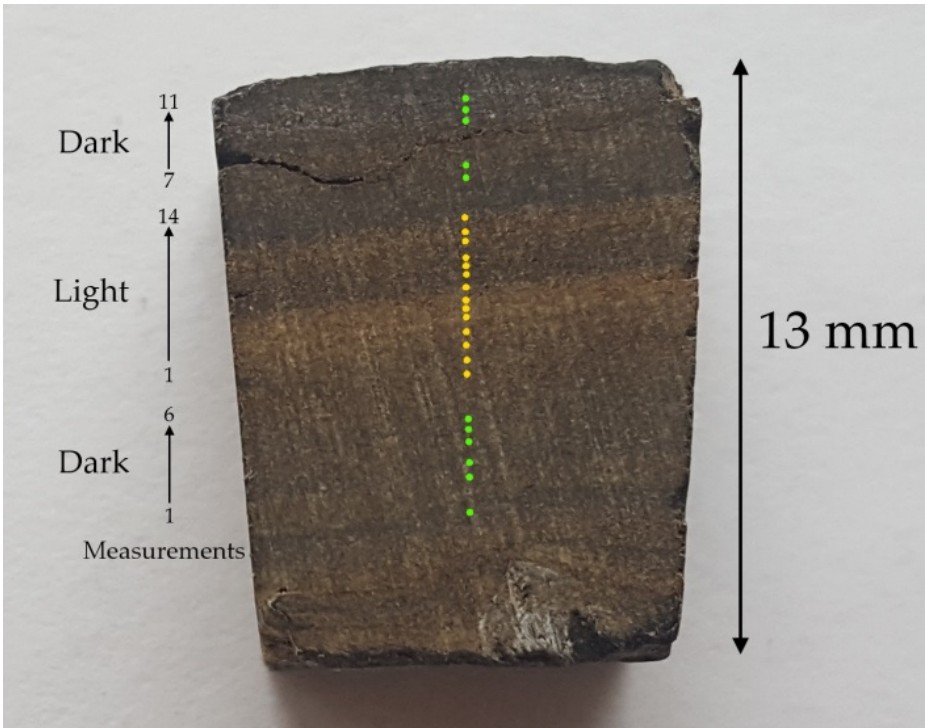

**Figure 7.** Photo of the Bazhenov Formation oil shale sample. Dark and light laminae, and linear measurement areas are highlighted on the images.

We expected that the studied lighter layers of the Bazhenov Formation sample would contain a high amount of fluorescing alginite components, which are typical for type I kerogen. An earlier study of such layers by pyrolysis and FTIR analysis [47] showed the predominance of aliphatic chains bonds in the structure of organic matter over aromatic structures, which correspond to a high H/C ratio (up to 1.70). So, in our study, we have chosen the major C-H stretching (2920–2928 cm$^{-1}$) band for content evaluation of aliphatic fragments and the C-C ring stretching (1620–1630 cm$^{-1}$) band for aromatic fragments. However, it should be noted that the aliphatic peak intensity associated with immature organic matter decreases as the kerogen is converted to bitumen, oil, and gas [31]. The aromatic carbon that dominates in thermally mature samples exhibits its most prominent peaks in the same region of the spectra as the broad peak for carbonate C=O stretches (1100−1600 cm$^{-1}$). It is evident that the reliability of organic matter content prediction using FTIR spectra is lower than for samples of lower thermal maturity, which rely on the more distinct C-H stretches of aliphatic moieties (2800–3000 cm$^{-1}$). There is a possibility that prediction error may be a significant problem for more mature samples.

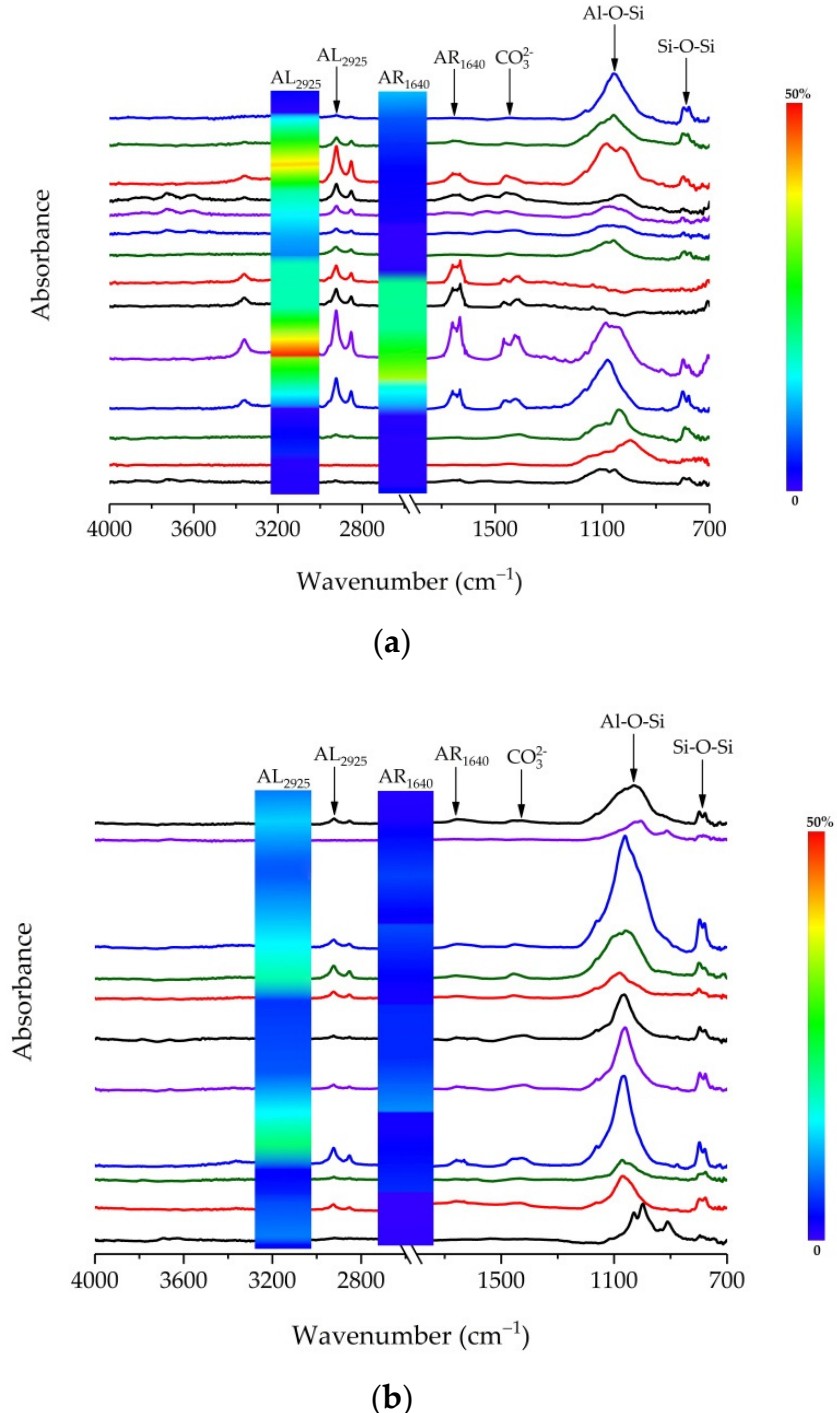

**Figure 8.** ATR spectra (FTIR-microscopy) and aliphatic and aromatic fragments content distribution plots for the Bazhenov Formation layered sample: (**a**) 14 measurements in light lamina; (**b**) 11 measurements in dark lamina.

The organic matter and mineral composition of the studied sample, estimated by the normalization method (sum of components normalized to 100%) using ATR spectra (Figure 8), is shown in Table 5.

**Table 5.** The evaluation of organic matter (OM) content and composition (aliphatic/aromatic ratio) and mineral content in the Bazhenov rock sample by FTIR microscopy.

| Lamina Measurement | | OM Content by Micro-FTIR, wt.% | | Ratio | Mineral Composition by Micro-FTIR, wt.% | | | OM Content by TGA, wt.% (Mass Loss up to 650 °C) |
|---|---|---|---|---|---|---|---|---|
| | | $AL_{2920–2928}$ | $AR_{1629–1640}$ | AL:AR | Clay | Carbonates | Quartz | Program 3 (Inert) |
| Light | 1 | 5.3 | 10.5 | 0.5 | 52.6 | 5.3 | 26.3 | - |
| | 2 | 8.0 | 2.0 | 4.0 | 66.0 | 2.0 | 22.0 | - |
| | 3 | 5.6 | 2.8 | 2.0 | 58.3 | 8.3 | 25.0 | - |
| | 4 | 22.8 | 15.2 | 1.5 | 39.1 | 7.6 | 15.2 | - |
| | 5 | 25.6 | 25.6 | 1.0 | 25.6 | 13.2 | 9.9 | - |
| | 6 | 36.4 | 48.5 | 0.8 | 0.0 | 15.2 | 0.0 | - |
| | 7 | 35.3 | 50.0 | 0.7 | 0.0 | 14.7 | 0.0 | - |
| | 8 | 20.0 | 4.0 | 5.0 | 52.0 | 0.0 | 24.0 | - |
| | 9 | 21.1 | 5.3 | 4.0 | 36.8 | 15.8 | 21.1 | - |
| | 10 | 33.3 | 4.8 | 6.9 | 42.9 | 0.0 | 19.0 | - |
| | 11 | 44.4 | 14.8 | 3.0 | 29.6 | 0.0 | 11.1 | - |
| | 12 | 35.2 | 9.9 | 3.6 | 42.3 | 0.0 | 12.7 | - |
| | 13 | 16.2 | 5.4 | 3.0 | 56.8 | 0.0 | 21.6 | - |
| | 14 | 4.3 | 0.0 | - | 71.7 | 0.0 | 23.9 | - |
| | Average $\pm \delta$ ($p = 0.95$) | **22.4 $\pm$ 7.7** | **14.2 $\pm$ 9.4** | **1.5 $\pm$ 1.2** | **41.0 $\pm$ 12.5** | **5.9 $\pm$ 3.7** | **16.6 $\pm$ 5.1** | **17.53** |
| | SD | **13.3** | **16.3** | **1.9** | **21.7** | **6.5** | **8.8** | - |
| Dark | 1 | 4.7 | 0.0 | - | 86.0 | 0.0 | 9.3 | - |
| | 2 | 10.7 | 5.4 | 2.0 | 58.9 | 5.4 | 19.6 | - |
| | 3 | 6.7 | 3.3 | 2.0 | 63.3 | 6.7 | 20.0 | - |
| | 4 | 11.2 | 4.2 | 2.7 | 61.5 | 5.6 | 17.5 | - |
| | 5 | 4.5 | 3.4 | 1.3 | 68.5 | 4.5 | 19.1 | - |
| | 6 | 4.7 | 3.1 | 1.5 | 67.2 | 6.3 | 18.8 | - |
| | 7 | 15.8 | 5.3 | 3.0 | 60.5 | 0.0 | 18.4 | - |
| | 8 | 17.8 | 4.1 | 4.3 | 61.6 | 0.0 | 16.4 | - |
| | 9 | 5.3 | 2.0 | 2.6 | 72.7 | 0.0 | 20.0 | - |
| | 10 | 0.0 | 0.0 | - | 81.8 | 0.0 | 18.2 | - |
| | 11 | 8.3 | 5.0 | 1.7 | 61.7 | 3.3 | 21.7 | - |
| | Average $\pm \delta$ ($p = 0.95$) | **8.2 $\pm$ 3.6** | **3.3 $\pm$ 1.3** | **2.1 $\pm$ 0.7** | **67.6 $\pm$ 6.1** | **2.9 $\pm$ 1.9** | **18.1 $\pm$ 2.2** | **12.69** |
| | SD | **5.3** | **1.9** | **1.0** | **9.1** | **2.9** | **3.2** | - |

The organic matter (aliphatic and aromatic fragments) and mineral content in the dark and light laminae of the shale sample have been evaluated using the major IR bands (Figure 8, Table 5): aliphatic C–H stretch (2920–2928 cm$^{-1}$), aromatic C-C ring stretch (1629–1640 cm$^{-1}$), carbonate (1408–1446 cm$^{-1}$), clay (995–1111 cm$^{-1}$) minerals and quartz (793–800 cm$^{-1}$). The color variations in the sample are reflected by changes in organic matter and mineralogy content along the length of the sample. Darker colored lamina contained lower aliphatic (8.2 $\pm$ 3.6 wt.%) and aromatic (3.3 $\pm$ 1.3 wt.%) fragments and carbonate minerals (2.9 $\pm$ 1.9 wt.%), and more clay minerals (67.6 $\pm$ 6.1 wt.%) than the lighter areas. Figure 8 shows the heterogeneous distribution of organic matter content (through aliphatic C-H stretch 2920–2928 cm$^{-1}$) for the Bazhenov Formation sample: from 4 to 44% in light and from 0 to 18% in dark laminae. Moreover, the evaluated average organic matter content by FTIR microscopy, calculated as a sum of aliphatic ($AL_{2920–2928}$) and aromatic ($AR_{1629–1640}$) contents, in light (18.1 wt.%) and dark (11.4 wt.%) laminae correlates well with the bulk TGA (inert atmosphere) results: 17.53 and 12.69 wt.%, respectively (Table 5). The aliphatic/aromatic content ratio (AL:AR) reflects the heterogeneous distribution of organic matter composition, particularly in light lamina: from 0.5 to 6.9. The average value of the AL:AR ratio in light and dark laminae is more than 1.5, which confirms the predominance of aliphatic chains bonds in the structure of organic matter over aromatic structures and the possible content of alginite in the studied laminae.

Despite general agreement between the averaged FTIR results and the TGA data on the organic matter content (Table 5), there are large standard deviations (SD) for

most values, particularly in light lamina, providing important insights into the extent of sample heterogeneity.

This approach can also be used to understand the scale of variations that are present within a reservoir and refine the geological model of the unconventional reservoir. The previous results reveal that bulk measurement for mudrocks associated with unconventional petroleum resource are not as useful as the micro-FTIR approach.

## 4. Conclusions

The content and chemical composition of organic matter of a series of the Bazhenov Formation kerogen and rock samples were examined by Fourier transform infrared spectroscopy (FTIR) in transmission and ATR modes. The organic matter content in the kerogen samples using transmission FTIR spectroscopy obtained by the internal standard method (from 31 to 52 wt.%) are in good agreement with those obtained by the normalization method (from 24 to 46 wt.%), as well as Rock-Eval pyrolysis (calculated using TOC value). ATR FTIR is more convenient for directly probing the sample without complex sample preparation and is useful for the characterization of mineral compositions and the evaluation of organic matter content in shale rocks, including the polished rock samples, which can be used for distribution analysis by FTIR microscopy. The use of ATR-IR spectra for a comparative assessment of the content of organic matter in kerogen is proposed and substantiated.

The approach with the removal of carbonate minerals in rock samples using 36% HCl to increase the intensity of the absorption bands of aliphatic and aromatic groups in the transmission spectra was used. This approach expands the capabilities of the IR spectroscopy method for studying the structure of organic matter directly in rocks without the stage of kerogen release. The organic matter content calculated from FTIR results as the sum of aliphatic and aromatic fragments values are in good agreement with that obtained by Rock-Eval pyrolysis and TGA (inert gas). Furthermore, it was shown that the AL:AR ratio could be used to study the chemical structure of organic matter in oil shales. FTIR spectroscopy also makes it possible to evaluate the content of minerals in rocks using the normalization method.

FTIR microscopy (ATR, normalization method) used for surface distribution analysis of the polished rock sample revealed the heterogeneity of the Bazhenov Formation sample organic matter and mineral composition. As expected, lighter colored laminae contained more aliphatic (up to 44 wt.%) and aromatic (up to 50 wt.%) fragments compared to darker laminae. The average value of the AL:AR ratio is more than 1.5, which confirms the predominance of aliphatic chains in kerogen structure over aromatic moieties, and the possible presence of alginite in the studied sample. The average organic matter content, calculated as a sum of aliphatic and aromatic contents, in light (18.1 wt.%) and dark (11.4 wt.%) laminae correlates well with the bulk TGA (inert atmosphere) results. It was shown that the heterogeneous distribution of organic matter content (through aliphatic C-H stretch) and composition (AL:AR ratio) for the Bazhenov Formation sample, particularly in light lamina, was from 4 to 44 wt.% and from 0.5 to 6.9 wt.%, respectively. Large variations in organic matter and mineral content within an interval may lead to more variability in production rates between wells; therefore, a better understanding of this variability could facilitate the improvement of probabilistic production models [48].

Thus, the main advantages of the considered approach for the study of oil shales by FTIR spectroscopy (transmission and ATR) and FTIR microscopy (ATR) are: easy sample preparation, and the possibility of semi-quantitative evaluation of the organic matter content (by normalization and internal standard methods) and composition (through the ratio of aliphatic and aromatic fragments), as well as surface analysis of the polished rock samples with subsequent mapping.

**Author Contributions:** Conceptualization, Y.P. and M.S.; methodology, Y.P., J.K., E.K. and E.L.; validation, Y.P., N.T. and J.K.; formal analysis, E.K., E.L. and N.T.; investigation, Y.P. and N.T.; resources, M.S., E.K. and Y.P.; writing—original draft preparation, Y.P.; writing—review and editing, M.S., E.K.,

J.K. and E.L.; visualization, Y.P. and N.T.; supervision, M.S. and Y.P.; project administration, Y.P.; funding acquisition, Y.P. All authors have read and agreed to the published version of the manuscript.

**Funding:** This work was supported by the Ministry of Science and Higher Education of the Russian Federation under agreement No. 075-10-2020-119 within the framework of the development program for a world-class Research Center.

**Data Availability Statement:** Not applicable.

**Acknowledgments:** The authors acknowledge the Government of the Khanty-Mansiysk Autonomous Okrug–Yugra and the Ministry of Science and Higher Education of Russian Federation.

**Conflicts of Interest:** The authors declare no conflict of interest. The funders had no role in the design of the study; in the collection, analyses, or interpretation of data; in the writing of the manuscript, or in the decision to publish the results.

## Appendix A

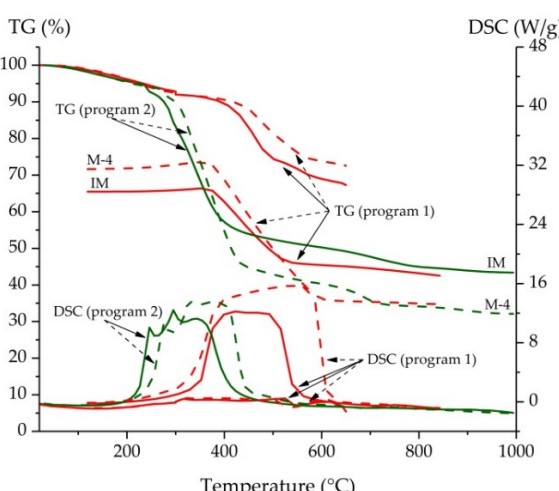

**Figure A1.** TG and DSC curves for kerogen samples. Green curves are for oxidizing atmosphere (Program 2); red curves for inert gas measurement followed by oxidizing atmosphere and for inert gas measurement (Program 1): IM—solid line, M-4—dashed.

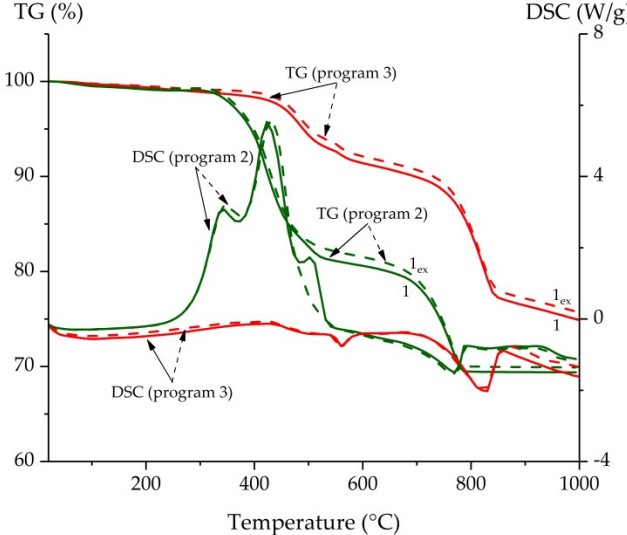

**Figure A2.** TG and DSC curves for rock samples. Green curves are for oxidizing atmosphere (program 2); red curves for inert gas measurement (program 3). Samples: before (1)—solid line; after ($1_{ex}$) bitumen extraction—dashed.

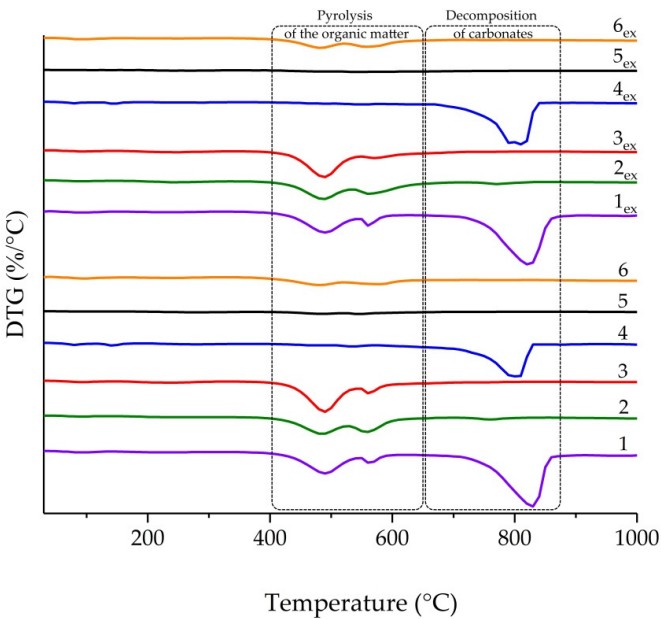

**Figure A3.** Comparison of DTG curves for rock samples before (1–6) and after ($1_{ex}$–$6_{ex}$) bitumen extraction (TGA measurement under inert atmosphere).

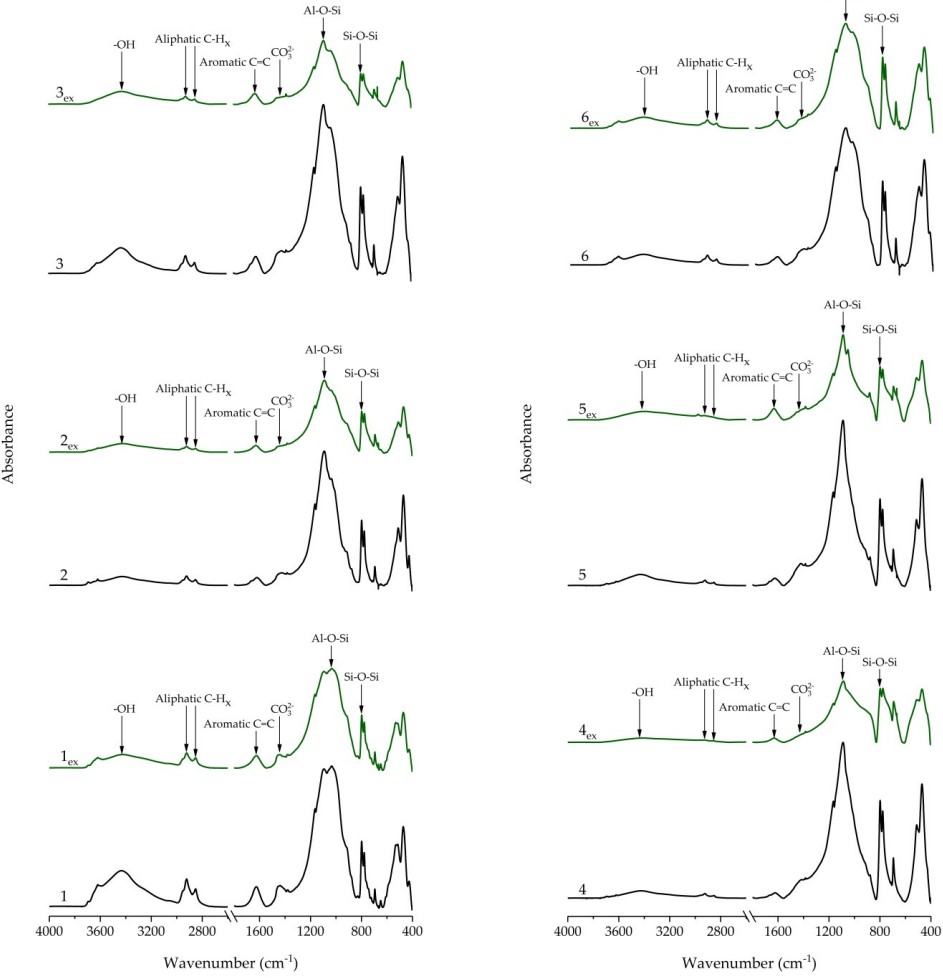

**Figure A4.** FTIR spectra in transmission mode of rock samples before (1–6, black) and after bitumen extraction ($1_{ex}$–$6_{ex}$, green) treated with 36% HCl (0.25% of powdered rock in KBr pellet).

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
