# Peer review of "Study of Organic Matter of Unconventional Reservoirs by IR Spectroscopy and IR Microscopy"

_geosciences, doi:10.3390/geosciences11070277_

Round 1
Reviewer 1 Report
The paper, dealing with the characterization of organic matter content and composition of unconventional reservoir with IR spectroscopy and IR microscopy, is well structured and written. It provides new insights on organic matter study trough IR methodologies. Methods are appropriate, as well as the interpretation of the results and figuring.
I have minor issues regarding:
Pag. 3 “The pyrolysis peak (Tmax), which reflects the thermal maturity of the kerogen, was between 477 and 544°C, that indicates kerogen of type II.”
Question: How the kerogen type can be determined with the Tmax index? Please clarify this concept or delete it.
Pag. 3 “The high Tex of pyrolysis in measurements of fine-grained Palaeozoic rocks was consistent with the FTIR results, which indicate the presence of aromatic hydrocarbons.”
Question: It is not clear to which samples the authors are referring. Please clarify.
Conclusions: The conclusions are a mixing of introduction, methodology, results, interpretations and discussions. The paragraph must be synthetized avoiding repetitions and highlighting only the new aspects of IR spectroscopy and IR microscopy on organic matter characterization, in comparison to that already know on this field.
References: the citations in the text could benefit of some missing papers:
Mastandrea, A., Guido, A., Demasi, F., Ruffolo, S.A., Russo, F. (2011). The characterisation of sedimentary organic matter in carbonates with Fourier-transform infrared (FTIR) spectroscopy. Lecture Notes in Earth Sciences, 131, pp. 331–342.
Guido, A., Mastandrea, A., Demasi, F., Tosti, F., Russo, F. (2012). Organic matter remains in the laminated microfabrics of the Kess-Kess mounds (Hamar Laghdad, Lower Devonian, Morocco). Sedimentary Geology, 263-264, pp. 194–201.
Author Response
Dear Editor,
We appreciate you and the reviewers for your precious time in reviewing our paper and providing valuable comments. The authors have carefully considered the comments and tried our best to address every one of them. We hope the manuscript after careful revisions meet your high standards. The authors welcome further constructive comments if any.
Below we provide the point-by-point responses (1-3). All modifications in the responses have been highlighted in red.
Sincerely,
Yuliya Petrova, PhD
petrova_juju@surgu.ru
Associated Professor, Director, Department of Chemistry, Institute of Natural and Technical Sciences
The Surgut State University of Russia
Please see the attachment.

Reviewer 2 Report
The manuscript titled “Study of organic matter of unconventional reservoirs by IR spectroscopy and IR microscopy” is a paper that would be of interest to the readers of Geosciences. The authors discuss the application of infrared spectroscopy for the characterisation of organic matter relevant to shale exploration and development. In my view, the article is of a good standard and the data was well presented. However, there is room for further improvements and below are some comments which I believe can be handled as a minor revision.
- The introduction contains an excellent review of the literature, but it does not provide any information on why the authors have taken a combined approach of using both the ATR and KBr methods. Some clarification is needed on why both methods were used in the paper?
- I am a little confused by the internal standard method (KSCN) that was used for the organic matter evaluation. Why was this material chosen and how was the sample/KSCN mixture prepared? Also, how was the data processed, which peaks/frequencies were used and did the authors use the peak intensity or peak area in their calculations? More information is needed on this in the experimental section. In addition, the normalization procedure that was used in the paper requires further information.
- The conclusion section has not been written very well and needs more attention. Currently as it stands it is too long and contains a lot of information that is already known. It needs to be shortened and made more relevant. I would like to see some statements regarding what are the learnings from the studies that have so far been undertaken?
- Some of the figures (figures 1 and 2) were difficult to see/read. Their size could be increased and the quality needs some improvement.
Author Response
Dear Editor,
We appreciate you and the reviewers for your precious time in reviewing our paper and providing valuable comments. The authors have carefully considered the comments and tried our best to address every one of them. We hope the manuscript after careful revisions meet your high standards. The authors welcome further constructive comments if any.
Below we provide the point-by-point responses (1-3). All modifications in the responses have been highlighted in red.
Sincerely,
Yuliya Petrova, PhD
petrova_juju@surgu.ru
Associated Professor, Director, Department of Chemistry, Institute of Natural and Technical Sciences
The Surgut State University of Russia
Please see the attachment

Reviewer 3 Report
The authors have applied bulk and micro FTIR methods to characterize organic and mineral properties of shale samples from an important petroleum formation in Siberia. Comparisons to data generated by other characterization methods show semi-quantitative analysis potential of the spectral interpretation technique for examining organic matter content, which utilized a combination of organic moiety peaks to estimate TOC and other organic indicators. The results show promise and the methods described in this case study are relevant to the petroleum geoscience literature on the applications of spectroscopy to source rock characterization. I have no technical issues with the manuscript but do have a few suggestions for minor corrections (typos) and changes to word choice.
Introduction, paragraph 2: After "marine shales" (line 7) insert "devoid of terrestrial organic matter input". Some marine shales post-evolution of land plants do contain vitrinite due to contributions from continental sources.
Page 5: in a few places "helding there" should be changed to "holding"
Figure 1 and other spectral figures: If the journal allows, please consider making the figures showing FTIR spectra larger to facilitate closer examination of kerogen and mineral features.
Page 10 and elsewhere: in several places "Appendix" is misspelled (Apendix)
Table 4 and in the text: when you mention "Silicate" minerals, is this meant to include feldspars or something else? I would also note that quartz is technically a silicate. Maybe change the Silicate category to "Other silicates"?
Page 15: I noticed one contraction (It's) in the manuscript. Not a major issue, but generally contractions aren't used writing for scientific journals (though it is kind of a stuffy, often unwritten rule).
Appendix: consider adding HAWK pyrograms (time or temperature vs. FID response).
Author Response

(The authors gave the same response as above.)
